# Review on Microbubbles and Microdroplets Flowing through Microfluidic Geometrical Elements

**DOI:** 10.3390/mi11020201

**Published:** 2020-02-15

**Authors:** Ana T. S. Cerdeira, João B. L. M. Campos, João M. Miranda, José D. P. Araújo

**Affiliations:** CEFT, Transport Phenomena Research Center, Chemical Engineering Department, Faculty of Engineering, University of Porto, Rua Dr. Roberto Frias, 4200-465 Porto, Portugal; anacerdeira@fe.up.pt (A.T.S.C.); jmc@fe.up.pt (J.B.L.M.C.); jmiranda@fe.up.pt (J.M.M.)

**Keywords:** microdroplets, microbubbles, microfluidic geometrical elements, coalescence, breakup, deformation, two-phase flows, contraction, constriction, T-junction

## Abstract

Two-phase flows are found in several industrial systems/applications, including boilers and condensers, which are used in power generation or refrigeration, steam generators, oil/gas extraction wells and refineries, flame stabilizers, safety valves, among many others. The structure of these flows is complex, and it is largely governed by the extent of interphase interactions. In the last two decades, due to a large development of microfabrication technologies, many microstructured devices involving several elements (constrictions, contractions, expansions, obstacles, or T-junctions) have been designed and manufactured. The pursuit for innovation in two-phase flows in these elements require an understanding and control of the behaviour of bubble/droplet flow. The need to systematize the most relevant studies that involve these issues constitutes the motivation for this review. In the present work, literature addressing gas-liquid and liquid-liquid flows, with Newtonian and non-Newtonian fluids, and covering theoretical, experimental, and numerical approaches, is reviewed. Particular focus is given to the deformation, coalescence, and breakup mechanisms when bubbles and droplets pass through the aforementioned microfluidic elements.

## 1. Introduction

Flow patterns are defined as the different types of spatial distribution of immiscible phases that develop as they flow simultaneously in a channel. These patterns are essentially dependent on the channel geometry, fluid physical properties, and flow rates of each phase. Furthermore, within each flow pattern, the channel geometry has a large impact on the behaviour of characteristic hydrodynamic features. This impact must be studied in detail, since the geometry is directly involved on the performance of practical applications, ranging from micro to macroscales.

Macroscale pipe networks comprise fluidic elements that normally process two-phase mixtures. Among the most common are expansions, contractions, and constrictions. These fluidic elements can be found in chemical reactors, gas-lift in oil wells, petrochemical plants, natural gas transportation in subsea pipelines, and power generation units [1]. As an example, the splitting of Taylor bubbles into two daughter bubbles, being caused by the passage through an expansion, is still scarcely studied experimentally [2], and there are only two published computational works of Taylor bubbles rising through expansions in pipe diameter [3,4]. Taylor bubbles/droplets are long and wide bullet-shaped entities—occupying almost all the cross section of a channel—that appear in one of the most frequent multiphase flow patterns, named slug flow. Despite the intrinsic complexity involved, the slug flow hydrodynamics in straight channels is a well-documented and well-studied subject [5], but the cases that involve other fluidic elements still need to be systematically studied. For the motion of droplets through a sudden/gradual expansion/contraction, several numerical and experimental works considered the flow through horizontal pipes, but they are mainly focused on pressure drop [6,7,8,9].

The heat and mass transfer can be significantly enhanced in a microsystem due to the high surface volume ratio. Currently, bubble/droplet-based microfluidics has been successfully applied in chemical and biological analysis [10,11], the synthesis of advanced materials [12], sample pretreatment [13], and the encapsulation of cells [14]. Droplets are liquid entities that flow in a immiscible liquid continuous phase, and bubbles are gas entities also flowing in a liquid fluid. The use of droplets as microreactors presents a lot of advantages when compared with single-phase microfluidics, such as: confinement of reactants or prevention of longitudinal dispersion, and cross contamination between samples [15]. The reduction of unwanted adhesion/absorption of the material confined in droplets at the microchannel walls, the possibility of varying, in each droplet the physicochemical conditions under which chemical or biological process develop, and the facilitated heat/mass transport due to the fast mixing promoted by droplets are other benefits [15,16]. The use of bubbles might be interesting to produce contrast agents in medical applications [17], as a source of reactants in the gas phase to promote reactions in liquid or solid (i.e., microchannel wall) phases [18], and to study in vitro gas embolisms [19].

Microfluidic devices (length scales lower than one millimeter) comprise microfluidic elements, analogous to their macroscale counterparts, to transport and distribute fluids, to promote mixing, reaction, and mass transfer. In the case of multiphase flows, these microfluidic elements also include expansions, contractions, constrictions, obstacles, and T-junctions that can be used to promote the formation, breakup, or coalescence of microdroplets and microbubbles.

Taylor microbubbles/droplets have garnered significant attention in various fields, including enhanced oil recovery, surface cleaning, medical purposes, and chemical and biomedical (such as drug delivery) engineering applications [20,21]. Good knowledge on two-phase flow hydrodynamics in millichannels and microchannels is important for the design, optimization, and control of structured microflow systems, since the physics governing bubble/droplet passing through microfluidic elements is really complex [22,23].

The present review addresses several aspects of handling bubbles/droplets in some of the most common microfluidic elements and the theoretical, experimental, and numerical tools that are used to tackle this problem. The scope of this review is mainly oriented towards the physics, experimental, and modelling approaches, and the designs of microfluidic elements for the control of deformable bubbles and droplets. A summary of the main experimental and numerical studies, together with their applications, is provided after a general overview of the theoretical background supporting the study of two-phase flows in microfluidic geometrical elements. The kinematics, fluid dynamics, and deformation dynamics of the bubbles/droplets are also discussed. Particular emphasis is given to the analysis of coalescence and breakup mechanisms that bubbles and droplets experience when they pass through microfluidic elements. With this review, the overall picture regarding two-phase flows in microfluidic geometrical elements is systematized.

## 2. Theoretical Background

Firstly, it is important to provide an overview of the main concepts that are involved in the flow of bubbles and droplets through microfluidic elements. Section 2.1 enunciates a description of the squeezing process in these fluidic elements, together with a quantitative evaluation of the droplet deformation and the channel confinement. The main types of microfluidic elements reported in the literature, with different geometries and arrangements, are addressed in Section 2.2. A dimensional analysis and the main operation parameters (e.g., pressure and transit time) of the problem are also discussed (Section 2.3).

### 2.1. Squeezing Process in Microfluidic Geometrical Elements

The continuous phase transports shear stresses and deforms the dispersed phase when flowing around a dispersed phase. The deformation of the bubble/droplet depends on the flow conditions, physical properties of the dispersed phase, and geometry of the microfluidic element that is involved (Section 2.2).

During a squeezing process through a microfluidic element, the bubble/droplet can take different shapes, varying from a parachute to a dumbbell shape. In the particular case of Taylor bubbles, several interface shapes can be observed by tuning the Reynolds (Re) and Capillary (Ca) numbers [22]. The deformed bubble/droplet assumes normally a bullet form while considering a Newtonian bubble or droplet and a Newtonian continuous phase. The deformation of a Newtonian bubble or droplet in a viscoelastic continuous phase induces, in most of the cases, an ellipsoid shape [24].

The deformation of the dispersed phase is commonly quantified by the Deformation Factor, also known as the Elongation Factor. Additionally, the bubble/droplet deformation is associated to a change in the geometry of the channel, so, to describe the degree of deformation in relation to the size of the contraction, a confinement parameter, known as the Confinement Factor, can also be adopted. When the dimensions of the channels are smaller than the diameter of the droplets/bubbles, the Deformation Factor is set as the ratio between the maximum length of the droplet/bubble during the deformation process and the un-deformed droplet/bubble diameter. The Confinement Factor is defined as the non-dimensional ratio between the diameter of the un-deformed droplet/bubble and the diameter of the constricted channel. Summaries of the most relevant analytical expressions that are available in the literature for evaluating the Deformation and Confinement Factors are presented in Table 1 and Table 2, respectively.

### 2.2. Microfluidic Elements

Microfluidic elements can exist in many forms, and so, a wide variety of geometries and arrangements have already been designed (see Figure 1). The most common types are constrictions (Figure 1a), expansions (Figure 1b), and contractions (Figure 1c). Other cases of interest involve the study of the droplet/bubble dynamics through obstacles (e.g., cylinders, squares) in microchannels (Figure 1d), and in T-junction channels (Figure 1e).

Within constrictions, the number of possible arrangements is high, and Table 3 summarizes the main types of constrictions that are frequently reported in the literature.

### 2.3. Operational and Geometrical Parameters

#### 2.3.1. Non-Dimensional Parameters

When passing through a microfluidic element, a bubble/droplet deforms and changes the dimensions. The dimension of the deformed bubble/droplet, $L_{de}$, depends on: the physical properties of the continuous (denoted by the index c) and the dispersed (index d) phases, namely their density, ρ, and viscosity, μ, as well as on the surface tension between both of the phases, σ; velocity of the continuous phase, $U_{c};$ geometrical parameters describing the bubble/droplet, given by its length (before deformation), L; and, widths of the channels in different regions, $w_{1}$ e $w_{2}$. When working with microfluidic devices, the interfacial forces become more prominent than gravitational forces, which tend to be negligible. For this reason, the acceleration due to gravity does not influence the dimension of the deformed droplet and will not be considered in the following dimensional analysis.

Dimensional analysis, which was based on the Buckingham theorem, leads to the following relation for the bubble/droplet deformation:(1)$$\frac{L_{de}}{D}=f\left(\frac{\rho_{d}}{\rho_{c}},\frac{\mu_{d}}{\mu_{c}},\frac{U_{c}\mu_{c}}{\sigma },\frac{w_{1}\rho_{c}U_{c}}{\mu_{c}},\frac{w_{2}}{w_{1}},\frac{L}{w_{1}}\right)$$

This relations is equivalent to say that the deformation, $L_{de}/D$, is a function of the density ratio, $\rho_{d}/\rho_{c}$, viscosity ratio, $Χ=\mu_{d}/\mu_{c}$, Capillary number of the continuous phase, ${\mathrm{Ca}}_{c}=U_{c}\mu_{c}/\sigma$, Reynolds number of the continuous phase, Rec=w1ρcUc/μc, and geometrical ratios $w_{2}/w_{1}$, and $L/w_{1}$. Thus, Equation (1) can be rewritten as:(2)LdeD=f(ρdρc,μdμc,Cac,Rec,w2w1,Lw1)

The density ratio has been shown to have a negligible effect on the velocity of Taylor drops at the macroscale [35], and its effect is not usually studied in microchannels. Based on the previous analysis, Table 4 summarizes the main dimensionless groups that characterize two-phase flows in small channels and their relevance.

In spite of the aforementioned, several other dimensionless groups may arise from the recombination of the main dimensionless groups (Table 5). For example, the Weber (We) and Laplace numbers (La) result from the combination between Reynolds (Re) and Capillary numbers (Ca). We can also mention the Deborah number (De), which is an useful non-dimensional parameter when working with viscoelastic fluids.

#### 2.3.2. Pressure

Pressure is an operational parameter that is used to control the flow of the dispersed phase through the continuous phase [41,42]. The critical pressure is a particularly relevant value, since it is defined as the maximum extra pressure that is required to squeeze a bubble/droplet through a microfluidic element (usually a constriction). This critical pressure can be described by the Young–Laplace equation, which relates the capillary pressure difference (ΔP), i.e., the pressure difference over an interface between two fluids, the surface tension, σ, and the principal radii of surface curvature, $R_{1}$ and $R_{2}$:(3)$$\Delta P=\sigma \left(\frac{1}{R_{1}}+\frac{1}{R_{2}}\right)$$

For a spherical bubble flowing in a circular narrow channel, Equation (3) can be rewritten as:(4)$$\Delta P=\frac{2\sigma \cos\theta }{R}$$
where R is the channel radius, and θ the contact angle.

The Young–Laplace equation is the most widely used design criterion in two-phase flow microfluidics, being utilized to define the interfacial equilibrium between two static fluids (e.g., capillary surface). However, its application has some limitations: (i) it is only valid in quasi-static cases (non-static situations may result in sharper curvature or backflow) [43]; (ii) in systems involving non-Newtonian or viscoelastic fluids, since the effect of viscosity variation is not taken in consideration [44]; (iii) it is not valid in the presence of non-uniform stresses; and, (iv) wetting behaviour is not accounted for when dealing with channels with sharp corners [45].

The pressure drop along the constricted channel is no longer linear due to the existence of the interface (Figure 2). The pressure drop along the constriction results from the contribution of three different factors: (i) the major pressure loss determined by Hagen–Poiseuille law, which linearly increases with the velocity of the phases; (ii) the minor pressure loss due to the contraction and to the expansion, which exhibits quadratic growth as a function of the velocity of the phases; and, (iii) the surface tension term, which dynamically changes with the droplet movement along the channel [42].

#### 2.3.3. Transit Time

The transit time (also called passage time or residence time) is the time that the bubble/droplet takes to move from the inlet to the outlet of a channel, and it is inversely proportional to the average velocity in the constricted channel [47]. This parameter can be calculated by the analytical expressions that are presented in Table 6. The transit time of a bubble/droplet passing through a constriction is influenced by the physical properties of continuous and dispersed phases, and by the bubble/droplet interactions with the confinement walls, by means of collision and friction [42,48]. Accordingly, it is dependent on three types of parameters: operational conditions (pressure, velocity, or Capillary number), physical properties (contact angle, surface tension, dynamic viscosity, or shear modulus), and structures (channel length, confinement radius, or height of the confinement) [42,49,50]. The dependence on the physical properties of the fluids can be used to evaluate the stiffness and viscosity of the dispersed phase [42,51].

## 3. Experimental Studies

In the last two decades, many microdevices, comprising several types of the aforementioned geometrical elements, have been designed and fabricated, due to a large development of microfabrication technologies. Some have been specifically designed to promote two-phase flows with different flow patterns and characteristics. For example, as the fluids flow through a microfluidic element, it might develop vortexes at sharp corners, leading to flow pattern and topological changes, and irreversible pressure losses. Hence, two-phase flows in the basic microfluidic elements need to be deeply understood to prompt new innovations in microfluidic devices for bubble/droplet transport, control, breakup, and coalescence. Several experimental studies involving four types of microfluidic elements—namely, expansions, constrictions, T-junctions, and obstacles—have been reported in the literature and will be discussed in this section.

### 3.1. Expansions

Expansions are often used to promote coalescence in microchannels. The coalescence process is usually divided into three stages: the acceleration and elongation of the trailing bubble/droplet; the trailing bubble/droplet catching up with the leading one; and, drainage, thinning, and rupture of the thin film between the two bubbles/droplets [53].

Mazutis and Griffiths [54] presented a microfluidic approach to selectively control droplet coalescence in an expansion section. Several types of oils (with and without surfactant) were used as continuous phase, and pure water or solutions of water-sodium chloride, water-hydrochloric acid, and water-sodium phosphate as dispersed phase. The authors found that the coalescence efficiency depends on the contact time between droplets and the interfacial surfactant coverage of the droplets. This process was independent of the Capillary number, coalescence angle, nature of dispersed and continuous phases, and type of surfactants [54].

Fu et al. [55] studied bubble coalescence in non-Newtonian fluids in a microfluidic expansion. Two different cases of bubble coalescence were observed: (i) the coalescence of two different bubbles along the expansion section; and, (ii) at the entrance of the expansion section, one bubble breaks into two different ones, and then the two daughter bubbles recoalesce. The transition between both cases can be controlled by changing the gas and liquid flow rates. In both cases, three different mechanisms for bubble coalescence were observed: in-line equal-sized bubbles coalescence, in-line unequal-sized bubbles coalescence, and parallel unequal-sized bubbles coalescence.

### 3.2. Constrictions

Unlike numerical works (Section 4.1), experimental studies involving constrictions are scarce. Only a few experimental studies have been made to analyse the process of a bubble/droplet squeezing through this kind of microfluidic element.

Harvie et al. [56] combined experimental and numerical methods to study the deformation of a viscoelastic droplet passing through a microfluidic cylindrical 4:1:4 constriction. This work is a very good example of a study that uses experimental data to assess the numerical results. Two-dimensional (2D) simulations were compared with the experimental results for a three-dimensional (3D) planar constriction. The authors observed a good agreement between experimental and numerical results. Particularly, a forked tail inside the constriction was observed in both of the methods. However, the simulations were unable to predict the encapsulation, observed in the experiments, of the continuous phase after passing the constriction. The authors attributed this problem to an inadequate mesh resolution.

Chai et al. [57] experimentally studied the effect of geometry on pressure drop and on two-phase (nitrogen gas and deionized water) flow patterns. They studied one straight microchannel and two microchannel comprising alternated expansions and constrictions. They observed slug flow, annular, or single-phase liquid flow patterns, depending on the flow rates of both phases. The authors applied a homogeneous flow model (i.e., single-phase flow equation, with suitably averaged properties [58]) and a separated flow model (i.e., the two-phase are artificially separated into two streams—Lockhart–Martinelli correlation [59]) to predict pressure drop. The separated flow model predicted the experimental pressure drop better.

### 3.3. T-junctions

Among experimental works involving microfluidic elements, the great majority were carried out while using T-junctions. Droplet/bubble formation in this type of configurations has been widely studied by theoretical analyses, experimental observations, and numerical simulations [60,61,62,63,64,65,66,67]. Furthermore, T-junctions are also useful to study the coalescence and breakup mechanisms. In this subsection, focus will only be placed on experimental works that involve the study of coalescence and breakup. Droplet formation will not be analysed, since it is a subject outside the scope of the present review.

#### 3.3.1. Droplet Coalescence

Microdroplets are frequently used in microfluidics as reactors to assure the confinement of reactions and respective products, and the digitalization of chemical processes. In digitalized microdroplet processes, the reactants are mixed through microdroplets coalescence. For this reason, geometrical elements that promote coalescence have been developed, and they are frequently reported in literature describing practical applications.

Christopher et al. [68] experimentally studied the effect of collision speed and Capillary number on the coalescence of droplets in a microfluidic T-junction. The working fluids were water-glycerol solutions (droplets) and silicone oil (continuous phase). The authors observed that, for two droplets to coalesce, the collision speed must be low enough, since droplets colliding at high speeds interact but do not merge, or one droplet can split the other into two smaller droplets. At low Capillary numbers, the droplets coalesced over the entire range of droplets sizes considered, and a critical Capillary number $\left({\mathrm{Ca}}_{crit}\right)$ was found at 0.02 (i.e., for ${\mathrm{Ca}}_{crit}>0.02$ droplets do not coalesce).

Wang et al. [69] studied microdroplet coalescence while using water and alcohol as working fluids and two types of junctions: T-shape and Y-shape junctions with different collision angles (60° to 180°). As visible in Figure 3, after droplets contact (Figure 3b), two different possibilities may occur: the two droplets may coalesce (Figure 3d) or separate (Figure 3f). On one hand, the time interval from the contact of two droplets to their fusion is mainly affected by droplet size and the physical properties of the working fluids. On the other hand, the time interval from the contact of two droplets to their complete separation is strongly dependent of the droplet collision angle. It was also shown that another way to enhance the microdroplet coalescence was by using lower droplet collision angles.

#### 3.3.2. Droplet and Bubble Breakup

At a T-junction bifurcation, a single droplet or bubble may not break, leaving through one branch of the bifurcation (Figure 4a), or break into two smaller ones (Figure 4b). The outcome depends on the droplet/bubble initial aspect ratio and the strength of the deformation that was induced by the collision between the droplet/bubble and the channel walls. The breakup in symmetrical T-junctions is normally used to improve the production rate of bubbles or droplets. When the two bifurcation branches do not have the same width, the T-junction is called asymmetrical. Asymmetrical T-junctions are used to further manipulate the size of bubbles or droplets.

Link et al. [70] employed a T-junction to passively break a droplet into two daughter droplets. The flow could be tuned to induce symmetric or asymmetric breakups, resulting in daughter droplets of equal or unequal sizes, respectively. The device can be used to obtain droplets with a specific average size and size distribution. Link et al. [70] also presented an analytical model to explain the transition between breaking and non-breaking in terms of the stretching of droplet in an extensional flow, based on the classical Rayleigh–Plateau instability (i.e., a liquid cylinder under the influence of surface tension is unstable if its length is as long as its base perimeter).

Leshansky and Pismen [15] developed a 2D theory for droplet breakup in a T-junction that was driven by pressure decrement in a narrow gap. The authors found that the normalized critical droplet extension, l/w (l is the deformed droplet length and w the width of the microchannel), depends on the Capillary number. The authors argued that the squeezed neck of a droplet shows an arc shape, and the dependence of l/w on the Capillary number was $l/w\sim {\mathrm{Ca}}^{-1/5}$, with these findings being in good agreement with a previous study conducted by Link et al. [70].

Garstecki et al. [71] and Julien et al. [16] experimentally studied the mechanism of droplets breakup in a microfluidic T-junction at small Capillary numbers: 0.00008<Ca<0.008, and 0.0003<Ca<0.27, respectively. Garstecki et al. [71] found that the dynamics of breakup are dominated by the pressure drop across the droplet as it passes through the T-junction, and not by shear stresses. When the droplet fills almost the entire cross-section of the channel, this pressure drop results from the high resistance to the flow of the continuous phase in the thin film separating the droplet from the microchannel walls. Julien et al. [16] studied two liquid-liquid systems (fluorinated oil droplets in deionized water and water droplets in hexadecane) over a wide range of flow rates and fluid viscosities, and they observed three different behaviours: non-breakup, breakup with tunnelling (there is a visible gap between the droplet and the walls before breakup), and breakup with permanent obstruction (droplets keep obstructing the T-junction before breakup).

Fu et al. [72] and Lu et al. [73] conducted too studies similar to [16] for gas-liquid systems. Fu et al. [72] studied the breakup regime map of nitrogen bubbles in glycerol-water solutions, in the presence of a surfactant, for Capillary numbers ranging between 0.001 and 0.1. By changing the gas and the liquid flow rates, the authors observed four different flow patterns: breaking with permanent obstruction, breaking with partial obstruction, breaking with permanent gaps, and non-breaking. The experimental results suggest that the bubble breakup in a T-junction is similar to the droplet breakup [15,16,70,71]. Lu et al. [73] analysed the evolution of the gas-liquid interface (nitrogen-turpentine oil) during the breakup of bubbles, and observed the same four patterns. The breakup was irreversible and fast when the normalised minimum width of the bubble neck was less than the critical value (0.5–0.6); otherwise, the breakup was reversible and slow.

Wang et al. [74] and Ziyi et al. [75] studied the partial obstructed breakup of bubbles in microfluidic junctions. Wang et al. [74] used an asymmetric T-junction, and they concluded that this process can be divided into three stages: squeezing, transition, and pinch-off, according to the evolution of the minimum width of the bubble neck, $w_{m}$. This evolution can be described by a power-law relationship between $w_{m}$ and time. It was also observed that the critical neck width of the breaking bubble is related to the Capillary number (0.0004≤Ca≤0.006) and the length of the original bubble (2.74≤l/w≤4). Very recently, Ziyi et al. [75] reported a study about the partially obstructed breakup of bubbles in microfluidic Y-junctions. In Figure 5, it is possible to see the effects of the channel angle (30° to 150°), Capillary number, and bubble length on the bubble rupture process. The authors found that the curve of the bubble neck during the breakup is nearly a parabola, rather than the aforementioned circular arc that was proposed by Leshansky and Pismen [15].

In a study similar to the one condicted by Ziyi et al. [75], Chen et al. [76] analysed nitrogen-water two-phase flow splitting at microchannel junctions with different branch angles (30° to 150°). The test range covered slug, slug-annular, and annular flow patterns. The authors found that the phase split at the junction depends on the flow pattern at the inlet. For slug flow at the inlet, the dominant phase in both outlet branches is the gas phase. On the other hand, for annular flow at the inlet, the dominant phase at the outlet branches is the liquid phase. Furthermore, for all types of inlet flow patterns, the fraction of liquid taken off through the branches (i.e., the ratio between the liquid mass flow rate of the branches and the liquid mass flow rate at the inlet) did not decrease with an increasing branch angle.

### 3.4. Obstacles

As aforementioned, the presence of obstacles in microchannels constitutes another relevant geometrical element, and some important experimental studies that are related with their interaction with flowing droplets can be found in literature.

Protière et al. [77] studied the breakup of long viscous droplets passing through a circular obstacle partially blocking the microchannel. The authors observed that, when a droplet moving above a critical velocity collides with an obstacle, it breaks into two uneven droplets. Below the critical velocity, the droplet only bypasses the obstacle through one of the gaps. Therefore, for a viscosity ratio equal to 8 (Χ=8), droplet will break for Capillary numbers above 0.02.

Zaremba et al. [78] studied, experimentally and numerically, the passage of a droplet through a groove. Hexadecane and surfactant were used as continuous phase, and distilled water colored with ink as the dispersed phase. The numerical simulations were carried out while using the volume-of-fluid (VOF) methodology implemented in ANSYS Fluent (ANSYS, Inc., Canonsburg, PA, USA). The contact angle was set to 180° (i.e., the droplet did not touch the walls of the microchannel). The coupling of experimental and numerical techniques provided a detailed description of the interactions between the two immiscible liquid phases that were altered by the geometry of the groove. The droplet divides into two parts in both experimental and numerical studies, with one of them being immobilized before the groove.

## 4. Numerical Studies

Numerical modelling can play a crucial role on increasing researchers’ ability to understand complex two-phase flows in microfluidic elements. Modelling must deal with interactions between bubbles/droplets of the dispersed phase, between the dispersed phase and the continuous phase, and between the dispersed phase and the walls of the microfluidic elements. Numerical methods must be able to properly track the interface to fulfil these objectives, including the evolution of the contact line at the wall surface. The main outputs of the numerical modelling are the evolution of the shape of the dispersed and continuous phases, and the velocity and pressure fields. These data may deliver important insights into optimum design of microfluidic devices.

In numerical simulations of two-phase flows, the interface tracking methodologies applied must describe interface shape evolution with high level of detail. Additionally, the appearance of small gaps between the dispersed phase and the microchannel wall require high local mesh densities, which is also numerically challenging. Wörner has also discussed these issues [79]. This author compared different numerical methods that were implemented in commercial and open source software that have already been used to study this kind of processes, and identified the advantages and benefits of the numerical approaches that are available to the research community. This section is focused on the main findings of numerical works involving two-phase flow in the main types of microfluidic elements (i.e., constrictions, contractions, T-junctions, and obstacles).

### 4.1. Constrictions

As aforementioned, constrictions are one of the most common examples of microfluidic elements, and there is a large number of numerical studies that involve this geometrical element. Furthermore, these works regard systems with Newtonian fluids and non-Newtonian fluids, so, they are addressed below separately since the subjacent principles are different.

#### 4.1.1. Newtonian Fluids

Harvie et al. [30,56,80] studied droplet deformation in a 4:1:4 axisymmetric microfluidic constriction applying the volume-of-fluid method. The authors analysed the effect of Reynolds number (Re) and surface tension strength—S=1/(We+Ca)—on droplet deformation while using low viscosity Newtonian droplets and a high viscosity Newtonian continuous phase (viscosity ratio, Χ, equals 0.001) [80]. For high surface tension strength, droplets break into slugs within the constriction (Figure 6a). For low surface tension strength and moderate Reynolds number, the droplets form thin filaments, and remain with this shape after passing the constriction (Figure 6b). When surface tension strength is small and the Reynolds number is low, the droplet assumes the form of a “string of sausages”, due to the influence of large amplitude instabilities on the droplet surface (Figure 6c).

Park and Dimitrakopoulos [81] studied the passage of elastic capsules and isolated droplets through constrictions. The elastic capsules had a spherical undisturbed shape and an elastic membrane enclosing a Newtonian fluid with a viscosity λμ, with μ being the viscosity of the continuous phase and λ the viscosity ratio between phases. The authors studied a range viscosity ratios between 0.01 and 5 (0.01<λ<5) and a range of Capillary numbers between 0.01 and 0.1, for both capsules and droplets. In this numerical study, an interfacial spectral boundary element (SBE) method for membranes, as developed by the same authors [82], was used to describe the movement of the elastic capsule. The authors utilized their fully-implicit time-integration spectral boundary element algorithm [83], as well as their membrane spectral boundary element algorithm by imposing surface-tension interfacial conditions, to determine the droplet dynamics [84]. The numerical results show that both droplets and capsules assume an elongated shape with a maximum length and a minimum height in the contraction region, changing to a flattened parachute in the expansion region. Therefore, the deformation of the droplet/capsule is only related to the hydrodynamic effects that were introduced in the flow by the constriction [81]. However, while the droplet velocity consistently decreases with decreasing viscosity ratios, the capsule velocity is insensitive to the viscosity ratio. Nevertheless, the behaviour of capsules (regardless of the viscosity ratio) corresponded better to that of very viscous droplets.

Zhang et al. [43,45] developed two relevant numerical work, where the deformation of a droplet passing through a microfluidic constriction was computed while using the VOF methodology, implemented in the commercial software ANSYS Fluent. The authors studied the influence of channel geometry (e.g., circular, square, rectangular, and triangular) on the pressure field and droplet deformation. On one hand, for circular, square, and triangular channels, the pressure starts to increase before the droplet enters the constriction, and then decreases after optimum inlet velocity. On the other hand, for rectangular channels, the authors did not verify the existence of an optimum velocity, and the pressure continuously decreases with the increase of the inlet velocity [43]. Zhang et al. [45] presented analytical expressions to calculate the critical pressure on the four mentioned cross-sectional shapes. The circular cross-section provided the highest critical pressure and, in decreasing order, being followed by the square, triangular, and rectangular channels. Additionally, it was shown that the use of a rectangular cross-section with a high aspect ratio might result in droplet breakup at high speed values.

Zhang et al. [47] also developed an analytical model to predict the pressure that is required to squeeze a viscous droplet (1<Χ<50) through a circular constriction. The model is based on five assumptions: (i) low Reynolds number; (ii) axisymmetric geometry; (iii) elasticity, wall-adhesion, and bending elasticity of the droplet negligible; (iv) one-phase in the radial direction; and, (v) spherical droplet in the curvature change part. For low Reynolds (Re=1.8) and Capillary (Ca=0.0072) numbers, an analytical model was able to accurately predict the squeezing pressure obtained numerically by solving the Navier-Stokes equations.

#### 4.1.2. Non-Newtonian Fluids

Khayat [85] presented a 2D numerical study of two-phase flow of viscoelastic fluids in a hyperbolic convergent-divergent constriction, while using the boundary element method (BEM) and the linear Oldroyd-B constitutive model. The droplet and the continuous phase were assumed to be incompressible, with a viscosity ratio equal to 3. The author concluded that, under small deformation rates, the fluid elasticity tends to enhance the droplet deformation. In a subsequent paper, Khayat et al. [86] examined the effects of shear and elongation on a droplet deformation in the same geometry. As in the previous study, the flow was simulated while using the BEM method. Both, the dispersed phase (droplet) and the continuous phase, could be either Newtonian or Maxwell fluids. The droplet deformation is different in these two cases: a high-viscous Newtonian droplet slowly deforms at the beginning, but then the process accelerates; a viscoelastic droplet initially deforms quickly, but then the deformation rate tends to slow down.

More recently, the contributions to the understanding of the dynamics of non-Newtonian droplets in microfluidic elements have increased, with several numerical studies being performed on axisymmetric channels [30,87] and on 2D planar constrictions [24,56].

Harvie et al. [30] focused on two types of liquid-liquid systems flowing through constrictions: droplets of a shear thinning fluid dispersed on a Newtonian fluid and droplets of a Newtonian fluid being dispersed on a shear thinning fluid. In both cases, the Carreau model described the viscosity of the shear thinning fluid. In the first case, the authors studied low-viscosity Newtonian continuous phases ($10^{-3}<\mathrm{Ca}\;<\;10^{3}$) and observed that the droplets deformation was similar to that of Newtonian droplets with a viscosity that was equal to the average viscosity of the shear thinning liquid. In the second case, the authors observed that droplet deformation was smaller than in the first case. Harvie et al. [56] also experimentally studied the deformation of a viscoelastic droplet surrounded by a low-viscosity Newtonian liquid flowing through a thin 3D planar constriction. The viscoelastic effects were described by the Oldroyd-B model. The results were compared with the numerical results obtained for a 2D constriction (already mentioned in Section 3.2). The numerical model employed captures the deformation process well, with special emphasis on the forked tail that is characteristic of a viscoelastic droplet inside a constriction. The authors also observed the presence of an elastic stress inside the droplet, which might cause the entrainment and subsequent encapsulation of continuous phase fluid within the droplet as it leaves the constriction.

Chung et al. [24,46,88] studied the effect of viscoelasticity (Deborah number between 0.4 and 2.2) on droplet dynamics in a 5:1:5 planar constriction while using the finite element-front tracking method (FE-FTM). Again, the Oldroyd-B model described the viscoelastic fluid. According to the fluids characteristics, two different types of deformation were observed [24]. On one hand, a Newtonian droplet in a viscoelastic continuous phase results in an “ellipsoid-like drop”, and, on the other hand, a Newtonian droplet and a Newtonian continuous phase results in a “bullet-like drop”. In a subsequent paper, the same authors investigated the excess of pressure ($\Delta p^{+}$) that is required to move the droplet through the constriction [46]. The excess of pressure was roughly proportional to the viscosity ratio (Χ), and inversely proportional to the Capillary number (Ca). The excess of pressure was strongly linked to the film thickness ($\delta_{f}$) between the interface of the droplet and the walls in the thinner region of the microchannel; this film thickness increases with the Capillary number. Thus, a decrease in the Capillary number results in a decrease in the film thickness and an increase of the excess of pressure. Chung et al. [88] found that droplet deformation depends on the Capillary number, with the effect being more pronounced for smaller viscosity ratios. The authors also studied the formation of bilateral symmetric vortices in the droplet and found that circulation intensity inside the droplet increases with decreasing viscosity ratios.

### 4.2. Contractions

A microfluidic contraction has a simple geometry, but the flow has strong and distinct regions of extensional and shear strains, which are able to considerably deform the shape of a droplet. Several numerical studies have been done regarding droplet dynamics in contraction microchannels, assuring a good level of knowledge on this subject. Nevertheless, the great majority of these numerical works involve 2D models, which may present some limitations.

Khayat et al. [89] used a boundary element method to investigate the effect of contraction geometry, droplet initial shape, and fluids rheology on droplet deformation. The geometry of the contraction was characterised by the convergence ratio. Two convergence ratios were studied, three and ten. The authors investigated the influence of the viscosity ratio (Χ=0.5, 4, 10) on the deformation of circular and elliptic droplets positioned upstream to the contraction. For circular droplets and Χ=0.5, the droplet deforms at the contraction inlet, while, for Χ=10, the droplet behaves like a rigid solid particle. Similarly, the deformation of elliptic droplets decreased as Χ increases, being larger due to the elongated initial shape of the droplet. As expected, for the same flow rate, the deformation was larger for the channel, with the convergence ratio equal to 10.

Christafakis and Tsangaris [25] analysed the influence of the Capillary, Reynolds, and Weber numbers and viscosity ratio on the droplet dynamics in a 2D contraction microchannel while using the level-set method (LSM). These authors simulated systems with three or four consecutive droplets, and the droplet shapes, their deformation and migration compare well with literature data. The values of the non-dimensional numbers were chosen to avoid the breakup or coalescence of the droplets (Ca between 0.1 and 1, Re=20, Χ=4, and We between 2 and 20). The Taylor deformation factor (Section 2.1) was used to describe the geometrical state of the droplets and to evaluate their deformation.

As aforementioned, 3D models must be used when dealing with the deformation of droplets in non-circular microchannels, since the droplet curvature and the hydrodynamics is not accurately described by 2D models. Accordingly, Hoang et al. [38] performed 3D simulations, together with a theoretical analysis of the droplet dynamics in a planar microchannel contraction. Both of the phases were assumed to be incompressible Newtonian fluids, and inertial effects were neglected. A volume of fluid method implemented in the commercial software ANSYS Fluent was used, and the surface tension was treated as a body force by using the continuum surface force model, to track the interface [90]. The authors identified three different regimes: trap, squeeze, and breakup (Figure 7). They proposed simple theoretical models, which were found to agree well with the numerical data, in order to elucidate the transitions between each regime. Droplet deformation and retraction along of the microchannel were analysed.

### 4.3. T-junctions

The breakup mechanism of droplets on T-shaped junctions is also a topic that has been addressed by computational approaches and, hence, some numerical studies are available in the literature.

Bedram and Moosavi [91] studied the breakup of non-uniform droplets through an asymmetric T-junction while using the volume-of-fluid (VOF) methodology. The T-junction comprised one inlet and two outlets with different dimensions. The authors also developed an analytical theory in the limit of the thin-film approximation. In this theory, for low Capillary numbers, they assumed that the pressure drop along the continuous phase mainly occurs in the thin film between the top wall of the microchannel and the droplet. While using the analytical solution, the authors calculated the smallest film thickness as a function of the Capillary number and of the ratio between the dimensions of the two outlet microchannels. A good agreement between analytical and numerical data was obtained, namely for the critical droplet length as a function of the Capillary number and for the velocity of the droplet as a function of the average velocity of the continuous phase. Smaller droplets can be obtained by increasing the Capillary number and, in this way, it is possible to decrease the breakup time and increase the pressure drop.

Mora et al. [23] developed a 3D numerical work about breakup dynamics of an isolated Taylor droplet in a rectangular T-junction microchannel while using a volume-of-fluid based solver from an open-source CFD code (OpenFOAM^®^). S-CLSVOF was the algorithm used in this study, which is a method that combines the VOF (volume-of-fluid) and LS (level-set) methods. On one hand, VOF provides accurate mass conservation and, on the other hand, LS an exact representation of the interface. A new mathematical correlation was proposed for the dimensionless neck thickness (i.e., minimum width of the droplet at junction) of the droplet versus Capillary number, dimensionless time, and dimensionless droplet length. In the breakup regime, the authors identified two different behaviours: non-obstructed breakup and obstructed breakup. The first one can also be divided into two subcategories, named breakup with permanent tunnel (i.e., a gap is clearly visible between the droplet and the channel), and breakup partially obstructed.

### 4.4. Obstacles

In the literature, there are also some numerical approaches available regarding the dynamics involved in the passage of a droplet through obstacles in microchannels. Chung et al. [92] presented an interesting study, both numerical and experimental. The FE-FTM was employed to study the deformation of a 2D droplet for interface tracking and a calculation of interfacial tension. Oil was considered as continuous phase, and water as dispersed phase. Four different cases were studied: a droplet flowing through a confined microchannel with a single cylinder obstacle (Figure 8a), a single square obstacle (Figure 8b), successive cylinder obstacles (not shown), and an array structure of cylinders (Figure 8c). The authors studied the effects of the obstacle shape, droplet size, and Capillary number (Ca) on droplet dynamics. For droplet size control, the cylinder obstacle showed better performance than the square obstacle. Low Ca droplets and enough distance between cylinders were necessary to have an independent/efficient wetting of the cylinders surfaces. Another relevant finding was that droplet size did not play a significant role on the droplet dynamics.

## 5. Perspectives and Challenges for the Future

Several researchers have studied, from a fundamental perspective, microfluidic elements to manipulate bubbles and droplets, covering a wide range of geometries and fluid properties. However, on the one hand, the experimental studies do not cover the full range of cases of interest and the numerical studies are frequently standalone studies without validation with experimental results. On the other hand, much remains to be done to apply the results from fundamental studies to the design of new devices for practical applications.

Two-phase flows in microchannels are affected by the surface properties of the walls of the channel. The surface properties influence the contact angle when the bubble/droplet gets in contact with the wall. This effect has been neglected in the literature on two-phase flows through microfluidic elements. From an experimental perspective, the control of the surface properties at the microlevel during microfabrication procedures is required to develop devices that can manipulate droplets through the local contact angle. On the numerical perspective, the numerical methods must deal with contact angle, which is a difficult subject that requires numerical innovations and experimental validation.

Microfluidic elements can be used in practical applications, such as microparticle fabrication, chemical reactors, mixers, and particle processing and separation. Most of the main functions that are required for these applications, such as droplet breakup and coalescence, have already been studied at the element level. However, practical applications require the scale-up of the device through parallelization. New studies are needed to understand how microfluidic elements can be used to conduct large scale production. Additionally, new applications need to be identified, exploring the surface properties effects and the combined effects of different elements. For example, droplets can be split or redirected by surface patches with different hydrophobicities and two-phase flows in complex networks can be used to improve the mixing. These geometries can be optimized by computational methods, i.e., mixing, droplet size, size distribution, and pressure drop can be improved applying topological optimization.

## 6. Conclusions

In the last years, microstructured devices played an important role in the technological development of several areas, including medicine, biotechnology, cosmetics, food, materials, pharmaceutics, and energy. Two-phase flows and, consequently, the presence of bubbles and droplets is recurring in such devices. Recently, with the rapid development of microfabrication technologies, many new elements and topologies of microstructured devices (constrictions, contractions, expansions, obstacles, or T-junctions) have been designed and manufactured. Since these elements affect the flow, shape, breakup, and coalescence of bubbles and droplets, playing with their geometrical features can be crucial in attaining different types of goals in practical scenarios. The systematization of the data published about the flow of droplets/bubbles in such elements is the main motivation of this review. Theoretical, numerical, and experimental analysis are presented, covering gas-liquid and liquid-liquid flows, with non-Newtonian and Newtonian fluids.

The deformation of bubbles/droplets passing through a microfluidic element depends on the physical properties and flow rates of dispersed and continuous phases, and on the type of confinement involved. For a specific element, the Reynolds and Capillary numbers of both phases and the viscosity ratio are the main dimensionless numbers governing the deformation. Several analytical expressions available in the literature are presented to calculate the Deformation Factor of bubbles/droplets passing through microfluidic elements.

The most relevant experimental studies involving bubbles/droplets coalescence and/or breakup in expansions, constrictions, T-junctions, and obstacles are reviewed. Expansions are often used to promote bubbles/droplets deformation and coalescence. Symmetrical T-junctions are useful for promoting coalescence, but also breakup, in order to improve the production rate of bubbles or droplets. There are four possible flow patterns in the droplet/bubble breakup in T-junctions: breaking with permanent obstruction, breaking with partial obstruction, breaking with permanent gaps (i.e., a gap is clearly visible between the droplet and the channel), and non-breaking. Asymmetrical T-junctions are applied to further manipulate the size of bubbles or droplets. Constrictions can be used to fragment bubbles and droplets, and they have the potential to be applied in microparticle production. Obstacles can be used to produce complex two-phase flows, promoting the bubble/droplet breakup and chaotic flow in both phases. Combined in complex arrangements, obstacles could be used to promote the mixing of reactants in the continuous phase.

In numerical simulations of a droplet/bubble moving through microfluidic elements, the reliability of the interface tracking methodology applied is a key component due to the high level of detail that is required to describe the interface shape evolution. Different numerical methods are presented, such as the volume-of-fluid (VOF), the spectral boundary element (SBE), the boundary element method (BEM), the finite element-front tracking method (FE-FTM), the level-set method (LSM), and the S-CLSVOF method.

## Figures and Tables

**Figure 1 micromachines-11-00201-f001:**
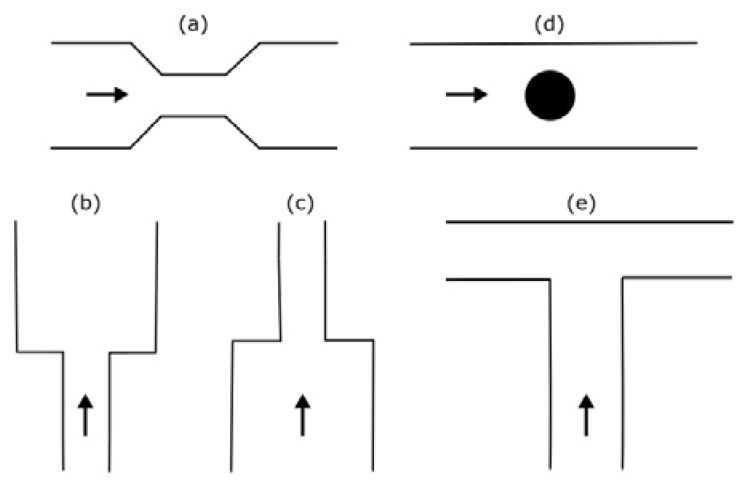
Schematic representation of the most common microfluidic elements: (**a**) constriction; (**b**) expansion; (**c**) contraction; (**d**) obstacle; and, (**e**) T-junction.

**Figure 2 micromachines-11-00201-f002:**
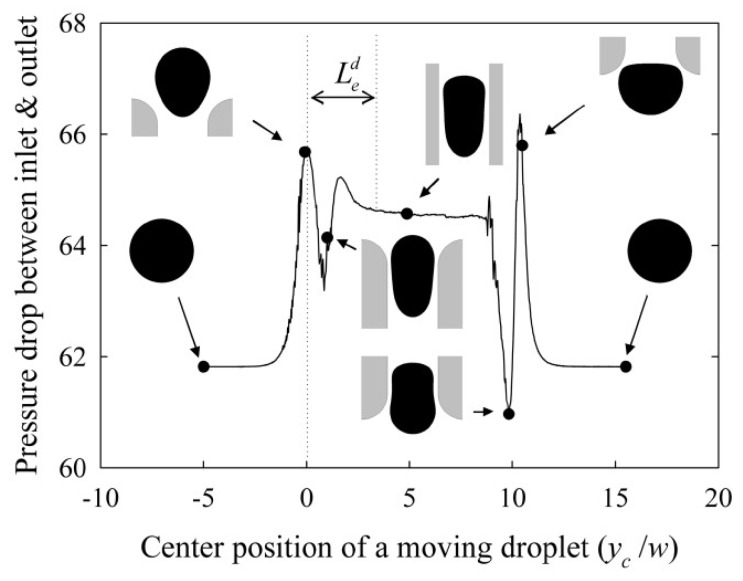
Pressure drop in a constriction between inlet and outlet with Χ=1 and Ca=0.1. $L_{e}^{d}$ is the entry length (i.e., the moving distance of the droplet since it enters the constriction until the pressure becomes constant), w the constriction microchannel width, and $y_{c}$ the centre position of the moving droplet along the microchannel. Reproduced with permission from [46].

**Figure 3 micromachines-11-00201-f003:**
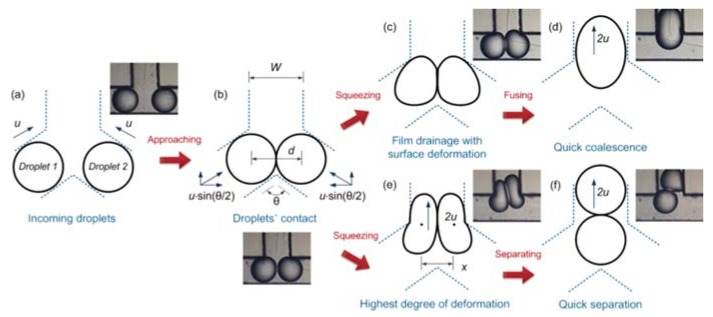
(**a**) droplet flow; (**b**) droplets contact; (**c**) squeezing; (**d**) fused droplet; (**e**) strongest deformation; and, (**f**) separated droplets. Reproduced with permission from [69].

**Figure 4 micromachines-11-00201-f004:**
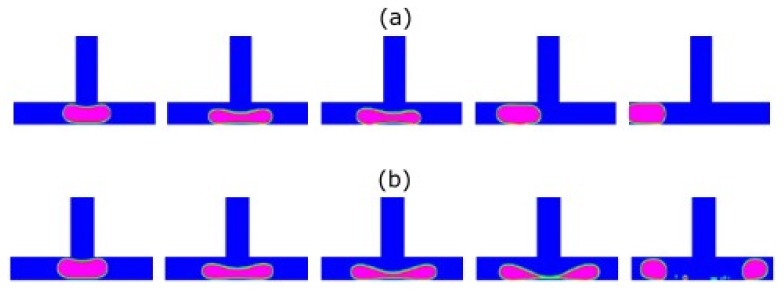
Droplet shapes from two-dimensional (2D) numerical simulations with time progressing from left to right in a symmetric T-junction: (**a**) non-breaking droplet; and, (**b**) breaking droplet. Reproduced with permission from [15].

**Figure 5 micromachines-11-00201-f005:**
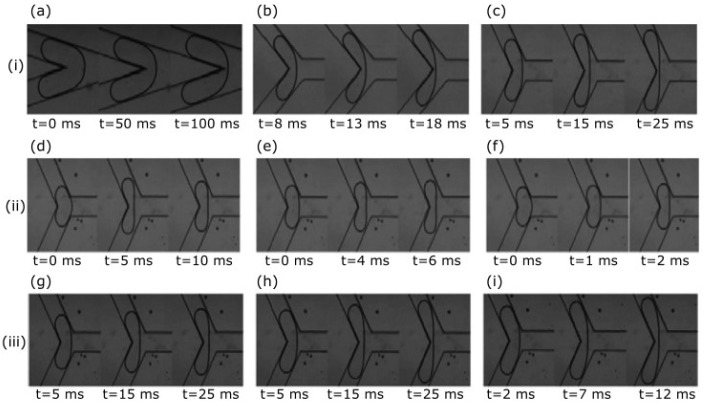
(i) Effect of the channel angle on bubble breakup: (**a**) 30°; (**b**) 90°; (**c**) 150°; (ii) Effect of the Capillary number, Ca, on bubble breakup: (**d**) Ca=0.021; (**e**) Ca=0.025; (**f**) Ca=0.029; (iii) Effect of the bubble length, L, on bubble breakup: (**g**) L=1.03 mm; (**h**) L=1.24 mm; and, (**i**) L=1.42 mm. Adapted with permission from [75].

**Figure 6 micromachines-11-00201-f006:**
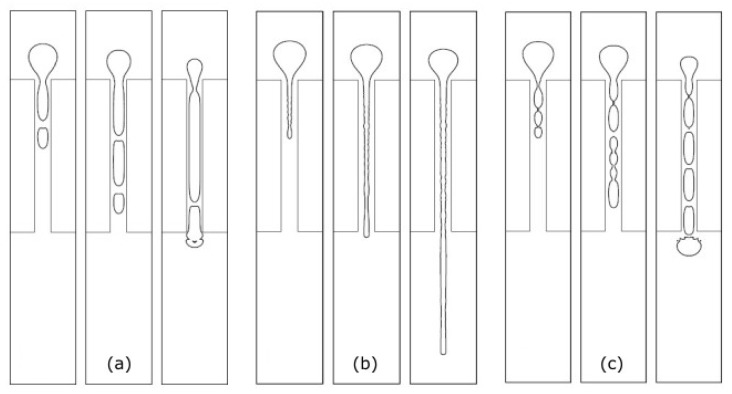
Low-viscosity droplet deformation behaviour in constrictions for Newtonian fluids with X = 0.001: (**a**) Re = 2.13 × 10^−3^ and S = 2.13 × 10^1^; (**b**) Re = 6.59 × 10^−1^ and S = 3.97; and, (**c**) Re = 6.76 × 10^−3^ and S = 6.71. Adapted with permission from [80].

**Figure 7 micromachines-11-00201-f007:**
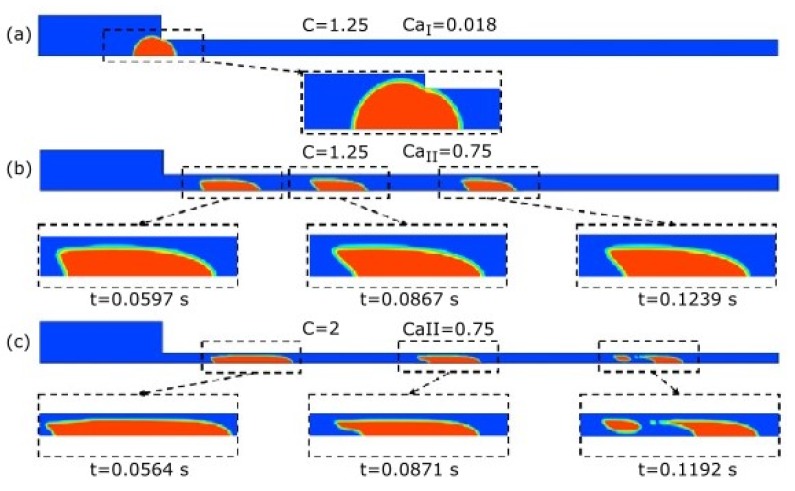
Droplet cross-section at the symmetric plane viewed from trop: (**a**) trap regime; (**b**) squeeze regime; and (**c**) breakup regime. C is a non-dimensional number representing the contraction level (C=D/W, where D is the droplet diameter, and W the contraction microchannel width). In the thicker part of the microchannel (i.e., before the contraction), the inlet velocity is employed as the characteristic velocity, and the Capillary number is denoted as ${\mathrm{Ca}}_{I}$, while the average velocity in the thinner region of the microchannel (i.e., after the contraction) is used to define the Capillary number ${\mathrm{Ca}}_{\mathrm{II}}$. Adapted with permission from [38].

**Figure 8 micromachines-11-00201-f008:**
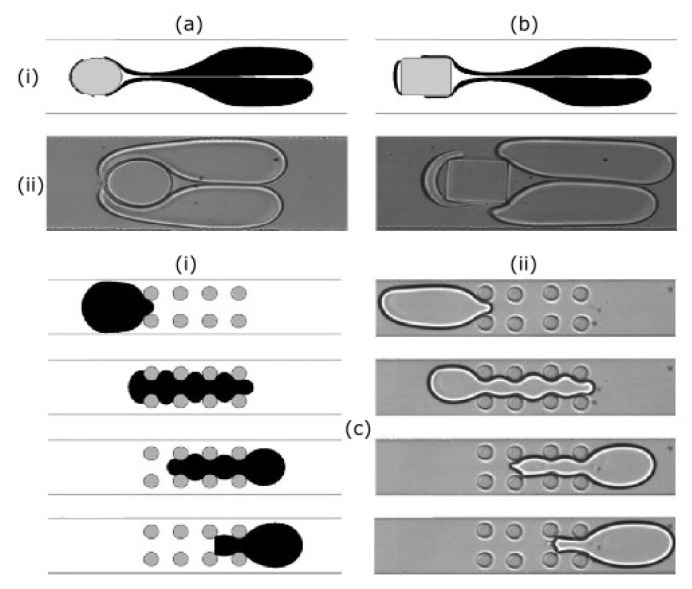
Droplet passing through the obstacles: (**a**) single cylinder; (**b**) single square; and, (**c**) array of cylinders; (i) numerical results; (ii) experimental results. Adapted with permission from [92].

**Table 1 micromachines-11-00201-t001:** Analytical expressions available in the literature for the Deformation Factor of a droplet/bubble.

Deformation Factor	Comments	References
$\;\;\frac{L-B}{L+B}\;\;$	Taylor deformation factor: L: maximum length of the droplet/bubble during the deformation process; B: minimum length of the droplet/bubble during the deformation process.	[25]
$\;\;\frac{L}{D}\;\;$	L: maximum length of the droplet/bubble during the deformation process; D: un-deformed droplet/bubble diameter.	[26]
$\;\;\frac{P}{2\sqrt{\pi A}}\;\;$	P: imaged perimeter of the droplet/bubble; A: imaged area of the droplet/bubble.	[27]

**Table 2 micromachines-11-00201-t002:** Analytical expressions that are available in the literature for the Confinement Factor of the microchannel.

Confinement Factor	Comments	References
$\;\;\frac{A_{D}}{A_{H}}\;\;$	$\;\;A_{D}:$ cross section of the un-deformed droplet/bubble; $A_{H}:$ cross section of the constricted channel.	[28]
$\;\;\frac{D-d}{D}\;\;$	D: diameter of the droplet/bubble; d: diameter of the constricted channel.	[29]
$\;\;\frac{r}{r_{H}}\;\;$	r: un-deformed droplet/bubble radius; $r_{H}:$ hydraulic radius of the constricted channel: $r_{H}=\frac{W_{c}H}{W_{c}+H}$ $(W_{c}$ is the width of the constricted channel, and *H* the height of the constricted channel).	[27]

**Table 3 micromachines-11-00201-t003:** Constriction channel geometry designs.

Constriction Geometry		Design Purpose	References
Straight	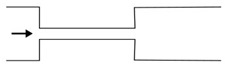	Droplet breakup	[30]
Tapered/ conical	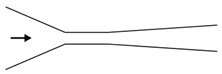	Droplet breakup	[27]
Sinusoidal/ cosine	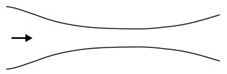	Oil trapping, droplet breakup	[31]
Double-bend	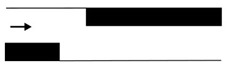	Provides deformation and rotation	[25]
Saw-tooth-shaped	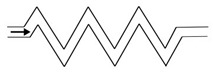	Directional transportation by periodic half-open capillary channels	[32]
Hyperbolic channel	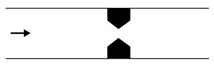	Capsule dynamics, encapsulate droplet	[33]
Shark teeth	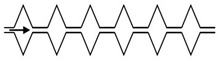	Provides rapid deformation	[34]

**Table 4 micromachines-11-00201-t004:** Main dimensionless numbers that governs two-phase flows in microscale.

Non-Dimensional Parameters		Function
Reynolds number of the continuous phase (Rec)	Rec=w1ρcUcμc	Ratio between inertial and viscous forces; Definition of the flow regime of the continuous phase and transportation process optimization; It is independent of the properties of the dispersed phase.
Capillary number of the continuous phase (${\mathrm{Ca}}_{c}$)	$\;\;{\mathrm{Ca}}_{c}=\frac{U_{c}\mu_{c}}{\sigma }\;\;$	Ratio between shear stress and surface tension; It is the most widely used non-dimensional parameter in a droplet squeezing process; Multiphase flow parameter deciding co-flow behaviour [36,37], flow regime [38], and passing/stuck [39].
Viscosity ratio between dispersed and continuous phases (Χ)	$\;Χ=\frac{\mu_{d}}{\mu_{c}}\;\;$	-

**Table 5 micromachines-11-00201-t005:** Dimensionless numbers sometimes used when working with two-phase flows.

Non-Dimensional Parameters		Function
Weber number (We)	We=ρU2w1σ=Re×Ca	Ratio between inertia and surface tension; Multiphase flow parameter useful in analysing droplets formation and thin-film flows.
Laplace number (La)	La=ρσw1μ2=ReCa	It is independent of external dynamics parameters, such as velocity, and only includes the properties of the fluids and geometric characteristics of the channel.
Deborah number (De)	$\;\mathrm{De}=\frac{\tau_{c}}{\tau_{p}}\;\;$	Rheology parameter quantifying viscoelastic effects [40]; $\tau_{c}$ is the relaxation time and $\tau_{p}$ the time of observation.

**Table 6 micromachines-11-00201-t006:** Analytical expressions available in the literature for the transit time of a bubble/droplet passing through a microfluidic element.

$\mathrm{Transit}\;\mathrm{Time}\;(t_{p})$	Comments	References
$\;\;t_{p}=\frac{V_{D}+V_{C}-2V_{0}}{\bar{U}A_{C}}\;$	$\;\;V_{D}$: volume of the droplet; $V_{C}$: volume of the constricted channel; $V_{0}:$ volume of the droplet inside the constricted channel at initial contact; $\bar{U}\text{:}$ average flow velocity in the constricted channel: $\bar{U}=\frac{1}{A_{C}}\int_{0}^{R}U2\pi rdr$ (*R* is the radius of the constricted channel); $A_{C}$: area of the constricted channel.	[47]
$\;t_{p}\propto {\left(\mathrm{Ca}-{\mathrm{Ca}}_{crit}\right)}^{-1/3}$	Flow-induced droplet squeezing through lattice of spheres; Ca*:* Capillary number; ${\mathrm{Ca}}_{crit}:$ Capillary number below which trapping occurs.	[52]

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
