# Peer review of "Review on Microbubbles and Microdroplets Flowing through Microfluidic Geometrical Elements"

_micromachines, 2020, doi:10.3390/mi11020201_

Round 1

Reviewer 1 Report

This is a well-constructed review of the current understanding of the transmission and modification of microbubbles and microdroplets flowing through microfluidic elements. The authors have concentrated on theoretical studies and experimental studies focussed on the detailed processes. As a result they have not considered much of the literature which is aimed at harnessing these effects as manufacturing processes. This is understandable, but it is worth noting that some of those papers do include quite detailed investigations (including using high-speed cameras to capture the processes going on at T-junctions -- see, for example, Combining microfluidic devices with coarse capillaries to reduce the size of monodisperse microbubbles, Jiang, X., Zhang, Y., Edirisinghe, M. and Parhizkar, M. (2016) RSC Advances 6:68, 63568-63577). The coverage of the paper is, however, made fairly clear in the introduction to this review. It might be helpful to state in the introduction what criteria the authors are using to distinguish microfluidics from fluidics, perhaps with some explicit length scales.

One further paper that might be mentioned is Experimental Investigation of Bubble Formation in a Microfluidic T-Shaped Junction, Zhang, Y and Wang, L (2009) Nanoscale and Microscale Thermophysical Engineering, 13:4, 228-242, DOI: 10.1080/15567260903276999.

The discussion throughout the review is clear and unbiased, and the writing is very good. One or two minor points (by line)
40 Although -> Despite
159 criteria -> criterion
284 breakink -> breaking
Some figures have been captured using a lossy format (JPEG?) which has blurred the text -- this is particularly apparent in Figure 3.

Overall this is a very informative and timely review, and I am happy to recommend publication: I would not insist on any changes.

Reviewer 2 Report

This manuscript presents an extensive review, although no original work, on the theory, computational work and experimental results for the formation and dynamics of bubbles and droplets in microscale fluidic devices. As such it could be a good resource for researchers that are looking into getting an understanding of where the field currently is and what are the various approaches used to study bubbles/droplets on these scales.

With that said, I believe that the work would be made more useful with the inclusion of the following:

The "Introduction" section should include a much more extensive discussion of why studying the formation and evolution of bubbles in microfluidic devices is important. What are specific applications that depend of bubbles/droplets for their functionality, or are hindered by the formation of bubbles? This is very important as it sets the context for the whole paper and the different experiments and simulations presented. The name of "bubbles" and "droplets" seem to be used interchangeably. If they are seen as the same thing then one nomenclature should be chosen. If not, then a definition distinguishing between the two should be specified early in the manuscript. For Tables 4 & 5 specify where applicable the common names for the non-dimensional parameters mentioned.  Some of the figures that are reproduced with permission, seem to be of poor quality (e.g. Figure 3 and 5). This will have to be replaced with higher resolution reproductions.
